# A Phase II Study of Denileukin Diftitox in Patients with Advanced Treatment Refractory Breast Cancer

**DOI:** 10.3390/vaccines13020117

**Published:** 2025-01-24

**Authors:** William R. Gwin, Lupe G. Salazar, James Y. Dai, Doreen Higgins, Andrew L. Coveler, Jennifer S. Childs, Rosie Blancas, Yushe Dang, Jessica Reichow, Meredith Slota, Hailing Lu, Mary L. Disis

**Affiliations:** 1Cancer Vaccine Institute, University of Washington, 850 Republican Street, Box 358050, Seattle, WA 98195, USA; lsalazar@uw.edu (L.G.S.); acoveler@uw.edu (A.L.C.); childj@medicine.washington.edu (J.S.C.); ydang@medicine.washington.edu (Y.D.);; 2Fred Hutchinson Cancer Center, 1100 Fairview Ave. N., Seattle, WA 98109, USA; jdai@fredhutch.org; 3Breastlink Medical Group, 230 S Main St, #100, Orange County, CA 92868, USA; 4Seattle Genetics (Pfizer), 21823 30th DR SE, Bothell, WA 98021, USA

**Keywords:** denileukin diftitox, regulatory T-cells, breast cancer, immunity, tumor microenvironment

## Abstract

**Background/Objectives**: Regulatory T cells (Treg) suppress immunity in the tumor microenvironment, are linked to poor prognosis across breast cancer subtypes, and suppress the cytolytic function of cytotoxic CD8+ T cells. Denileukin diftitox, a diphtheria toxin (DT)/IL-2 fusion protein, targets and depletes Tregs. This Phase II study aimed to assess the safety of denileukin diftitox and its effect on Tregs and tumor growth in patients with advanced breast cancer. **Methods**: This single-arm Phase II study of denileukin diftitox enrolled patients with refractory stage IV breast cancer. Patients received denileukin diftitox 18 mcg/kg/day IV for Days 1–5 every 21 days for up to six cycles. Toxicity was assessed using CTCAE v3.0 and tumor response was evaluated per RECIST criteria. Blood samples were collected to analyze Tregs by flow cytometry and anti-DT antibodies by ELISA. **Results**: Fifteen patients with stage IV breast cancer were enrolled. Four patients completed all planned denileukin diftitox infusions and achieved stable disease (27%, 95% CI [0.08, 0.55]). Two patients (13%) discontinued due to toxicity, and nine patients (60%) discontinued due to progressive disease. Eleven patients experienced at least one grade 3 or 4 adverse event. Although there was a general reduction in peripheral blood Tregs, the difference in CD4+CD25+FOXP3+ Tregs levels post-treatment was not statistically significant (*p* = 0.10). Six patients (40%) achieved ≥25% reductions in Tregs. A significant increase in anti-DT IgG antibodies was observed post-treatment (*p* < 0.005). **Conclusions**: Denileukin diftitox demonstrated moderate toxicity in this advanced breast cancer cohort. Denileukin diftitox modulated regulatory T cells. However, the majority of patients experienced disease progression in the study.

## 1. Introduction

The role of the immune system in breast cancer has gained substantial recognition in recent years [1]. Traditionally, the presence of tumor-infiltrating lymphocytes (TILs) has been used as an indicator of immune interaction with tumors, with TILs correlating with improved overall survival, particularly in basal-like and HER-2 enriched subtypes [2,3,4]. Recent evidence suggests that the quantity and functionality of cancer-specific immune responses, both local and systemic, are crucial for prognosis [5,6]. Specifically, CD8+ cytotoxic T cells have been linked to better survival outcomes, independent of tumor size, stage, or HER2 status [7]. Conversely, regulatory T cells (Treg), found in both the tumor microenvironment and peripheral blood, are associated with poor prognosis across all breast cancer subtypes, with high Treg infiltration reported in 67% of HER2-enriched and 70% of basal-like breast cancers [8,9].

The presence of CD4+ Th1 T cells and their secretion of IFN-gamma are critical for the activation and proliferation of CD8+ T cells, promoting a Type I immune environment that facilitates direct tumor cell lysis [10]. In contrast, a Type 2 immune environment, characterized by CD4+ (Th2) T cells and CD4+CD25+FOXP3+ Tregs, secretes cytokines that inhibit CD8+ T cell activation. The presence of CD25 + FOXP3+ Tregs is linked to decreased relapse-free survival and poorer clinical outcomes in breast cancer [11]. Therefore, strategies to enhance Type 1 immunity while suppressing Type 2 responses are essential for improving anti-tumor efficacy.

Denileukin diftitox (DD) is a fusion protein composed of diphtheria toxin with interleukin-2 (IL-2) that is specifically designed to target cells expressing the IL-2 receptor (IL-2R). Upon binding to IL-2R, DD is internalized through receptor-mediated endocytosis, selectively targeting cells that express the high-affinity IL-2R, which includes CD25 (IL-2Rα), CD122 (IL-2Rβ), and CD132 (γc)] [12]. Tregs constitutively express three IL-2R subunits, making them a target for DD. After being withdrawn from the market in 2014, a re-engineered formulation of denileukin diftitox with enhanced purity and bioactivity (denileukin diftitox-cxdl) received FDA approval in August 2024 for cutaneous T cell lymphoma (CTCL) expressing IL-2Ra (CD25). This approval was based on results from a Phase III trial that demonstrated an objective response rate of 36.2% (95% confidence interval [CI] = 25.0%–48.7%), including a complete response of 8.7% [13]. 

Treatment with DD for other tumor types has also shown reductions in regulatory T cells in ovarian cancer (significant reduction in mean Treg levels, *p* = 0.025) [14], melanoma (Treg reductions in all treated patients) [15], and renal cell carcinoma (26–76% reduction in Tregs) [16]. These studies reported adverse events consistent with the known side effect profile of DD, with the most common grade 3 or 4 adverse event being vascular leak syndrome.

Our group has previously demonstrated in pre-clinical breast cancer models that DD reduces Tregs in the tumor microenvironment (*p* < 0.03) and is associated with immune-mediated tumor rejection [17]. Building on these findings, a Phase II study was conducted to assess the safety and efficacy of DD in targeting breast cancer cells, reducing Tregs, and inhibiting tumor growth in patients with advanced refractory breast cancer.

## 2. Materials and Methods

### 2.1. Patient Population

After providing informed consent, patients with stage IV breast cancer were enrolled in this trial from October 2006 to August 2009 (NCT00425672). The study was approved by the University of Washington Cancer Consortium institutional review board. Eligibility criteria included progressive or relapsed disease, measurable disease, and Eastern Cooperative Oncology Group (ECOG) performance status of ≤2. Concurrent use of endocrine and bisphosphonate therapy was allowed. Enrolled patients received intravenous (IV) denileukin diftitox (ONTAK) 18 mcg/kg/day on Days 1–5 every 21 days for a total of 6 cycles. Dose reductions were allowed for toxicity. Tumor response was evaluated per RECIST. Immunologic monitoring of the peripheral blood Treg population and endogenous tumor antigen-specific T cell immunity occurred at baseline and after cycles 2, 4, and 6.

### 2.2. Study Design

In this Phase II, non-randomized, single-arm clinical trial, the primary objectives were to evaluate (1) the safety of denileukin diftitox (ONTAK) infusion and (2) the effect of denileukin diftitox (ONTAK) administration on peripheral blood Tregs. Toxicity grading was evaluated according to the CTEP Common Terminology Criteria for Adverse Events (CTCAE) v3.0. The efficacy of denileukin diftitox in depleting Tregs was defined as a decrease in peripheral blood Tregs (CD4+CD25+FOXP3+) by 25% with denileukin diftitox treatment. To evaluate the anti-tumor response of denileukin diftitox, imaging studies were performed to assess target lesions at baseline and after cycles 2, 4, and 6 of treatment. Responses were categorized into four groups: (1) complete response (CR), (2) partial response (PR), (3) progressive disease (PD), and (4) stable disease (SD). The rates of CR, PR, SD, and PD were calculated along with their 95% confidence intervals.

### 2.3. Evaluation of Regulatory T Cells

The levels of CD4+CD25+FOXP3+ regulatory T cells in the peripheral blood were measured by flow cytometry, as previously published [18]. Briefly, peripheral blood mononuclear cells (PBMCs) were collected, washed, and incubated with 10% normal mouse serum in PBS. The cells were then aliquoted and stained with appropriate conjugated monoclonal antibodies at 4 °C in the dark. The antibodies used included human CD3, CD4, CD8, CD14, CD19, CD25, CD28, CD56, CD45RA, CD45RO, and appropriate isotype controls (all antibodies purchased from BD PharMingen, San Diego, CA, USA). Stained cells were then acquired with an FACS Canto flow cytometer (BD Bioscience, San Jose, CA, USA) and analyzed with FlowJo software (version 9).

### 2.4. Evaluation of Anti-Diphtheria Toxin Antibodies

Firstly, 96-well plates (Falcon MicroTest 111; Becton Dickinson, Oxnard, CA, USA) were coated with 1 pg of diphtheria toxin antigen (List Biological Laboratories, Campbell, CA, USA) per well and diluted in 50 µL of phosphate-buffered saline (PBS). The plates were incubated for 16 h at 4 °C [19,20]. Following this incubation, the plates were blocked for 1 h with 100 µL of a gelatin solution containing 0.05% Tween 20 in PBS (PBS-Tween). After blocking, the plates were rinsed three times with PBS-Tween. Next, eight serial five-fold dilutions of each serum sample (100 µL per well) were added to the coated plates and incubated for 1 h at room temperature. Afterward, 100 µL of protein A-alkaline phosphatase (Boehringer Mannheim Biochemicals, Indianapolis, IN, USA) diluted in PBS-Tween was added to each well, followed by another 1 h of incubation at room temperature. The plates were rinsed three times with PBS-Tween and twice with PBS. Finally, 100 µL of substrate solution (pNPP; Kirkegaard and Perry, Gaithersburg, MD, USA) was added to each well, and the plates were incubated for 10 min. After incubation, the plates were washed again and developed as previously described [21,22,23]. Diphtheria toxin-specific IgG antibodies at each time point are reported in mg/mL.

### 2.5. Statistical Methods

The confidence interval of the response rate was computed by the Clopper–Pearson exact method. For calculating the overall change in Tregs after treatment, we selected the maximal percentage of Tregs achieved post-denileukin diftitox for each individual and compared this to that individual’s baseline Treg percentage (maximal post-denileukin diftitox Treg percentage − baseline Treg percentage = maximal Treg percentage change). These two variables, maximal Treg percentage and baseline Treg percentage, were then compiled from all evaluable patients and compared using an exact binomial probability test. The association between the change in peripheral Treg level and the clinical response was evaluated using Cox proportional hazard (coxph) modeling. Differences in median anti-diphtheria toxin IgG antibody levels between baseline and post-denileukin diftitox treatment were assessed using a two-tailed Wilcoxon matched-pairs rank. All statistical analyses were performed using GraphPad Prism 7.01 (GraphPad Software, Inc., San Diego, CA, USA) or Microsoft Office Excel 2013 (Microsoft Corporation, Redmond, WA, USA). For comparisons involving more than three groups, two-way ANOVA followed by Bonferroni’s post-test was used when two variables were present. *p* < 0.05 was considered statistically significant in all analyses.

## 3. Results

### 3.1. Patient Characteristics

Fifteen female patients with advanced, refractory breast cancer were enrolled in the study. Each received at least one cycle of denileukin diftitox (administered 18 mcg/kg/day on days 1–5 of a 21-day cycle,). Table 1 presents the patients’ demographic and baseline tumor characteristics. Ten of the fifteen enrolled patients (67%) had received four or more prior lines of systemic therapy in the metastatic setting.

### 3.2. Feasibility and Toxicity of Administering Denileukin Diftitox for Treatment Refractory Breast Cancer

Of the fifteen patients enrolled in the study, four completed all six planned cycles of denileukin diftitox (Figure 1). One patient completed four cycles, and another completed three cycles. Additionally, five patients completed two cycles of therapy, while four patients completed only one cycle of therapy. Of the 11 patients who did not receive all planned denileukin diftitox infusions, two discontinued the study due to toxicity, and nine stopped treatment due to disease progression. Blood samples for immune analysis were available from both the baseline and post-denileukin diftitox therapy time points in ten patients (67%).

Eleven patients reported at least one grade 3 adverse event. Of the toxicities that were deemed at least possibly related to denileukin diftitox therapy, 8% were grade 3 (Table 2). There was one grade 4 toxicity, lymphopenia; this toxicity was felt to be related to the study treatment and resolved within one week. The overall most common treatment-related adverse events, regardless of grade, included fatigue, hypoalbuminemia, elevated liver functions, electrolyte imbalances, and acute vascular leak (Table 2). A complete list of the adverse events can be found in Appendix A. Two patients (13%) discontinued the study due to study-related toxicities, which included either profound fatigue and weakness or persistent thrombocytopenia. Additionally, three patients (20%) required a 25% reduction in the dose of denileukin diftitox due to the toxicities of nausea, vomiting, fatigue, and diarrhea.

### 3.3. Clinical Response to Denileukin Diftitox Therapy in Advanced Breast Cancer

Patients received anywhere from one to a maximum of six cycles of therapy. The best clinical response observed was stable disease in four patients (27%, 95% Clopper–Pearson CI [0.08, 0.55]), with a median duration of 13 months (ranging from 4 to 22 months). Notably, three of these four patients with stable disease had metastatic triple-negative breast cancer (TNBC). No patients achieved a complete or partial response. Nine patients (60%) discontinued therapy due to progressive disease.

### 3.4. Denileukin Diftitox Therapy Modulated Peripheral Blood FOXP3+ T Cells

Six of the fifteen (40%) evaluable patients achieved a ≥25% reduction in their peripheral blood CD4+CD25+FOXP3+ T cells following treatment. The mean reduction in FOXP3+ T cells in these six patients was 56.0% ± 10.96%. Most of these patients (five out of six) experienced this reduction after completing two cycles of therapy, with the remaining patient showing a 25% reduction after four cycles. Three of the four patients with stable disease showed a >25% decrease in Tregs with denileukin diftitox treatment. Three of the six patients with progressive disease and post-treatment Treg levels showed a >25% decrease in Tregs.

In the combined analysis of all evaluable patients with baseline and post-treatment data, there was no significant difference in the levels of CD4+CD25+FOXP3+ T cells when comparing the baseline to the maximal observed Treg levels (either decrease or increase) after denileukin diftitox therapy (*p* = 0.10, one-sided) (Figure 2A).

### 3.5. Denileukin Diftitox Therapy Induced Anti-Diphtheria Toxin Antibodies

Among the evaluable patients, there was a significant increase in the levels of IgG anti-diphtheria toxin antibodies at week 6 compared to baseline, *p* < 0.005 (Figure 2B). Notably, 100% of the patients (9 out of 9) who were assessed at both baseline and week 6 (after two cycles of denileukin diftitox) showed an increase in anti-diphtheria toxin IgG by week 6. 

## 4. Discussion

Regulatory T cells (Tregs), a subset of CD4+ helper T cells, play a critical role in modulating immune responses within the tumor microenvironment by producing cytokines that suppress the proliferation of effector CD8+ T cells [24,25]. In both pre-invasive and invasive breast cancers, Tregs are significantly elevated compared to matched normal tissue (*p* < 0.05) [26]. In contrast, higher levels of intratumoral CD8+ T-cells are associated with improved breast cancer-specific survival (HR 0.55 95 % CI, 0.39 to 0.78 *p*  =  0.001) [7], while the presence of intratumoral Tregs is associated with a worse prognosis across all breast cancer subtypes [8]. In the adjuvant breast cancer setting, elevated Treg levels are linked to shorter disease-free survival (DFS) (HR = 3.13, 95% CI = 1.23 to 7.89) and overall survival (OS) (HR = 7.69, 95% CI = 3.43 to 17.23) [27]. Similarly, in metastatic breast cancer, progression-free survival (PFS) inversely correlates with the number of peripheral blood Tregs (AUC = 0.970, *p* = 0.004) [28].

Given the role Tregs play in inhibiting effector T cells and suppressing anti-tumor immunity, multiple agents with anti-Treg activity have been evaluated across different tumor types, including cytotoxic agents like cyclophosphamide [29] and immune interventions such as anti-CTLA-4 antibodies [30]. However, the outcomes of these systemic approaches have been mixed. In this Phase II study, treatment with denileukin diftitox led to stable disease in a subset of patients, but did not result in any partial or complete responses. Additionally, denileukin diftitox modulated Treg levels, but only in a minority of patients.

Previous clinical trials of denileukin diftitox in non-hematologic malignancies have also investigated its impact on peripheral blood Tregs, yielding conflicting results. Treg reductions have been reported in renal cell carcinoma (26–76% reduction) [16], ovarian cancer (significant reduction of mean Treg levels, *p* = 0.025) [14], colorectal cancer (Treg reduction in the majority of evaluable patients) [31], and melanoma (Treg reduction in all treated patients) [15]. However, two other melanoma studies have reported no significant decrease in Treg levels after denileukin diftitox treatment [32,33]. No prior studies have specifically evaluated the use of denileukin diftitox in metastatic breast cancer. To date, there has been no definitive evidence of clinical benefit associated with Treg depletion in denileukin diftitox-treated patients [31,33].

Several factors may explain the low clinical and Treg response rate observed in our study. One possibility is the presence of neutralizing antibodies against denileukin diftitox. Previous studies in patients with cutaneous T-cell lymphoma (CTCL) [13,34], chronic lymphocytic leukemia (CLL) [35], CD25-expressing lymphomas [36], and melanoma [33] have identified anti-diphtheria toxin antibodies in a modest number of patients at baseline, with most patients developing these antibodies after treatment [13,33,34,35,36]. Another potential factor is the dosing schema used in this study. Clinical trials with shorter dosing schedules and lower doses of denileukin diftitox demonstrated statistically significant suppression of CD4+CD25+ Tregs [14,15,16,31]. These findings suggest that a truncated administration schedule and lower dosage of denileukin diftitox per cycle might be more effective in reducing circulating Tregs while also minimizing toxicity, as compared to prolonged treatment at higher doses.

## 5. Conclusions

Denileukin diftitox therapy in patients with advanced breast cancer led to modest modulation of circulating Tregs, but also resulted in moderate toxicity within this cohort. The majority of patients experienced disease progression, with only a minority achieving disease stabilization. To potentially improve outcomes, future studies should explore alternative dosing schedules that may offer more consistent Treg depletion and more favorable toxicity profiles.

## Figures and Tables

**Figure 1 vaccines-13-00117-f001:**
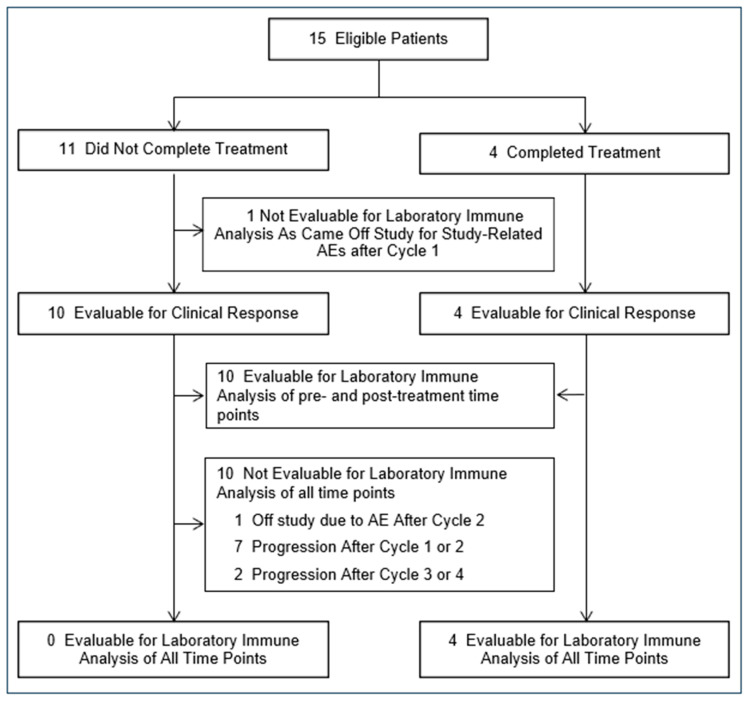
Patient flow chart.

**Figure 2 vaccines-13-00117-f002:**
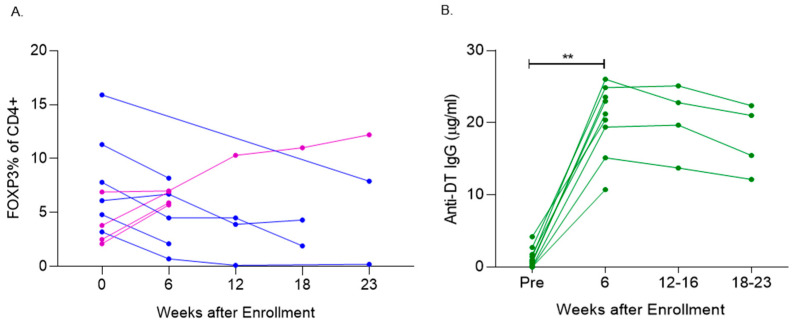
Denileukin diftitox (DD) therapy modulated peripheral blood FOXP3+ T cells and induced anti-diphtheria toxin (DT) antibodies. (**A**) Flow analysis evaluation of the % of peripheral blood CD4+ cells positive for FoxP3+ at the following study time points: baseline (week 0), week 6, week 12, week 18, and week 23. Five patients were evaluable with blood draws beyond week 6. Decrease in Tregs (Blue), increase in Tregs (Purple) Each line corresponds to an individual patient. (**B**) ELISA analysis of the change in diphtheria toxin (DT)-specific antibodies between baseline and post-treatment. Each line corresponds to an individual patient, and 100% (9/9) of pts who were evaluated at week 6 (after two cycles of DD) had an increase in anti-DT IgG from baseline ** *p* < 0.005.

**Table 1 vaccines-13-00117-t001:** Patient characteristics.

Characteristic		No. of Patients	%
Age (Median, Range)	58 (32–69)	15	100
Number of Disease Site(s)			
Bone, LN and soft tissue only (including skin)		6	40
Visceral		4	26
Mixed		4	26
Breast Cancer Subtype			
ER and/or PR+ *		6	40
HER2+ (overexpressing/amplified)		5	33
TNBC (ER/PR/HER2-)		4	26
Number of Prior Anti-Estrogen			
1–2 regimens		5	33
≥ 3 regimens		5	33
Number of Prior Cytotoxic Chemotherpay			
2–3 regimens		7	47%
≥4		8	
Number of Prior HER2 therapies			
1–2		4	26
Months from Last Therapy to Enrollment			
<1 month		8	
2–3 months		4	26
≥4 months		3	

* Two patient did not have outside pathology for ER/PR/HER2 testing, but had received prior anti-estrogen therapy and had not received anit-HER2 therapy, so they were grouped in ER/PR+ subtype.

**Table 2 vaccines-13-00117-t002:** Adverse events, including AEs judged to be possibly, probably, or definitely related to treatment with ONTAK.

Protocol 127 Adverse Events (n = 589)				
	Possibly, Probably, or Definitely Related		All AEs	
Most Common	No.	% of Related AEs	No.	% of All AEs
Fatigue	36	10	36	6
Hypoalbuminemia	27	8	27	5
Hypokalemia	26	7	31	5
ALT, SGPT elevated	22	6	33	6
Hypocalcemia	21	6	28	5
AST, SGOT elevated	19	5	34	6
Acute vascular leak syndrome	19	5	19	3
Nausea	15	4	22	4
Constipation	12	3	26	4
Hyperglycemia	4	1	21	4
AE Gradings				
1	242	67	416	71
2	87	24	142	24
3	30	8	30	5
4	1	0	1	0
5	0	0	0	0

## Data Availability

The datasets used and/or analyzed during the current study are available from the corresponding author upon reasonable request.

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
