# Peer review of "A Phase II Study of Denileukin Diftitox in Patients with Advanced Treatment Refractory Breast Cancer"

_vaccines, 2025, doi:10.3390/vaccines13020117_

Round 1
Reviewer 1 Report
Comments and Suggestions for Authors
This study aimed to assess the safety, efficacy, and impact on Tregs of denileukin diftitox in patients with advanced breast cancer.
I have a number of questions for the authors:
1. What was the expected clinical outcome the authors anticipated from the use of denileukin diftitox in this study?
2. Please elaborate on the expected mechanism of action of denileukin diftitox, which led the authors to anticipate its therapeutic effect in this setting.
3. Is it appropriate to evaluate the drug's efficacy in the entire patient group, given that not all patients received the same number of treatment cycles? I note that the variation in the number of cycles received could be considered substantial.
4. Given that the number of cycles was directly related to the treatment response, as therapy was discontinued upon disease progression, I believe the authors should reconsider the choice of the primary endpoint in this study. Instead of using the proportion of patients achieving stable disease or better, it would be more clinically relevant to use a parameter such as duration of clinical benefit. I can provide some relevant references to support this suggestion.
Sznol M. Reporting disease control rates or clinical benefit rates in early clinical trials of anticancer agents: useful endpoint or hype? Curr Opin Investig Drugs. 2010 Dec;11(12):1340-1
Delgado A, Guddati AK. Clinical endpoints in oncology - a primer. Am J Cancer Res. 2021 Apr 15;11(4):1121-1131. PMID: 33948349; PMCID: PMC8085844
Ultimately, I disagree with the authors' interpretation of the results. The most frequent outcome was disease progression, indicating that denileukin diftitox therapy was ineffective in this setting. However, the study's conclusions are limited by its small sample size.
Author Response
Please see the attachment. Also uploading edited manuscript with the revisions and a copy of the clinical trial protocol.

Reviewer 2 Report
Comments and Suggestions for Authors
The manuscript entitled “A Phase II study of denileukin diftitox (DD) in patients with advanced treatment refractory breast cancer” presents results on Tregs level obtained in the phase II study aimed to assess the safety of denileukin diftitox (DD) in stage III and IV breast cancer patients. DD is anticancer drug that was originally approved by the FDA in 1999 for the treatment of patients with persistent or recurrent cutaneous T-cell lymphoma (CTCL) that express the CD25 component of the IL-2 receptor on the surface of the malignant cells. DD was resubmitted and approved by the FDA on August 8, 2024. In this manuscript the authors presented only a part of the obtained data; however, it is crucial. But I have some important concerns.
Major concerns:
1. In the NCT00425672 clinical trial on clinicaltrials.gov it is stated that enrolled patients were those with male breast cancer. However, in the Results section, on line 149, the authors stated that “female patients” were enrolled. Please, clarify this in the manuscript.
2. The authors concluded that “Denileukin diftitox modulated Tregs and was associated with stable disease in a subset of patients”. However, the studied patient cohort was relatively small and amount, for which a stable disease was achieved with DD treatment included only 4 patients from a total 15 ones – about 27%.
3. The results obtained in this study showed that “Six patients (40%) achieved ≥25% 49 reduction in Tregs”. However, it is unclear how the decrease in Treg level correlated with the treatment response in each patient group.
Other concerns:
4. In the Introduction, it would be useful to provide data on phase II trials on advanced ovarian cancer [14], melanoma [15], and renal cell 85 carcinoma [16], for which DD has been shown to reduce peripheral blood Tregs; also, more importantly, provide discussion on toxicity of the drug in these cancer types.
5. Line 78, it should be “diphtheria toxin”. Table 1, make correction “ER+”. Lines 177-178, the authors stated “Six of the ten (40%)” – this should be corrected because 40% is six of fifteen (not ten).
6. Lines 114-116, Materials and Method. Provide short description of the “Evaluation of regulatory T cells”.
7. Figure 2 – does each line correspond to each patient? Clarify this in the figure legend.
Author Response

(The authors gave the same response as above.)

Round 2
Reviewer 1 Report
Comments and Suggestions for Authors
Based on the authors' responses to my queries, I contend that the study suffers from a paucity of empirical evidence, the study design is not sufficiently robust, and the manuscript falls short of fulfilling the essential criteria for publication.
Author Response
Reviewer 1 Comments 1: Based on the authors' responses to my queries, I contend that the study suffers from a paucity of empirical evidence, the study design is not sufficiently robust, and the manuscript falls short of fulfilling the essential criteria for publication.
|
Response 1: We acknowledge the reviewers’ comments regarding the content of this study. Ultimately as this trial is completed we cannot change the design nor the primary objectives.
The primary objectives as stated in the protocol of this Phase I-II clinical trial were to evaluate (1) the safety of denileukin diftitox in patients with advanced refractory breast cancer and (2) evaluate the effect of denileukin diftitox on peripheral blood Tregs and we report these in the manuscript. The study size was powered based on this later primary objective. The secondary objectives included clinical outcomes, levels of IL-2R in tumor tissue, sIL-2R in peripheral blood, and evidence of anti-tumor immunity. The analysis of the IL-2R in tissue and sIL-2R did not reveal significant association with denileukin diftitox treatment and as such were not reported in the manuscript.
|
Reviewer 2 Report
Comments and Suggestions for Authors
My concerns have been addressed. However, the authors provided responses to my Comments 2 and 3 in the Cover letter. How did they revised the manuscript according to these comments?
Author Response
Reviewer 2
Comments 1: My concerns have been addressed. However, the authors provided responses to my Comments 2 and 3 in the Cover letter. How did they revised the manuscript according to these comments?
Response 1:
We thank the author for this feedback. The edits to the manuscript to address comments 2 and 3 please see the manuscript lines 207 – 210 where the relationship between the decreased in Tregs and the clinical response was described in further details.